# A Look Under the Carpet of a Successful Eradication Campaign Against Small Ruminant Lentiviruses

**DOI:** 10.3390/pathogens14070719

**Published:** 2025-07-20

**Authors:** Fadri Vincenz, Maksym Samoilenko, Carlos Eduardo Abril, Patrik Zanolari, Giuseppe Bertoni, Beat Thomann

**Affiliations:** 1Veterinary Public Health Institute, Vetsuisse Faculty, University of Bern, 3097 Liebefeld, Switzerland; 2Institute of Virology and Immunology IVI, 3147 Bern, Switzerland; maksym.samoilenko@unibe.ch (M.S.); carlos.abril-gaona@unibe.ch (C.E.A.); 3Department of Infectious Diseases and Pathobiology, Vetsuisse Faculty, University of Bern, 3012 Bern, Switzerland; 4Clinic for Ruminants, Vetsuisse Faculty, University of Bern, 3012 Bern, Switzerland; patrik.zanolari@unibe.ch

**Keywords:** small ruminant lentiviruses, CAEV, MVV, VMV, eradication, emerging, reemerging

## Abstract

Small ruminant lentiviruses (SRLVs) are widespread and have a long co-evolutionary history with their hosts, namely sheep and goats. These viruses induce insidious pathologies, causing significant financial losses and animal welfare issues for the affected flocks. In Switzerland, in the 1980s, an eradication campaign was launched targeting these viruses, exclusively in goats, eliminating the virulent SRLV-B strains from the goat population, in which SRLV-B-induced arthritis was prevalent. Nevertheless, although they do not seem to induce clinical diseases, SRLV-A strains continue to circulate in Swiss goats. For this study, we contacted farmers who had animals testing positive for these strains during the census from 2011 to 2012 and visited six of these flocks, conducting serological, virological, and clinical analyses of the animals. We confirmed the absence of SRLV-B; however, we have detected SRLV-A in these flocks. Positive and negative animals lived in close contact for ten years and, except for a small flock of 13 animals, 7 of which tested positive, the transmission of these viruses proved inefficient. None of the positive animals showed any pathology attributable to SRLV infection. These encouraging results allowed us to formulate recommendations for the continued surveillance of these viruses in the Swiss goat population.

## 1. Introduction

Small ruminant lentiviruses (SRLVs) represent a complex species of the genus lentivirus of the Retroviridae family. The classical subdivision into a goat-specific caprine arthritis encephalitis virus (CAEV) and a sheep pathogen, Maedi-visna virus (MVV), has been abandoned since it was realized that these viruses do not possess a strict species specificity [1,2]. Some genus members appear to be restricted to goats, while others tend to be limited to sheep, but the majority are decidedly promiscuous [3,4,5].

This genetic complexity and variability are major obstacles to eradication campaigns, primarily because most of these endeavors were started before recognizing the complex biology of these viruses. In this respect, Switzerland paid the price for pioneering an eradication campaign in goats before the potential risk of SRLV transmission between sheep, excluded by this campaign, and goats was known. In the 1980s, Switzerland started an eradication campaign of CAEV that successfully eliminated the SRLV-B from the goat population and the clinical manifestations of this infection, i.e., carpal arthritis in adult goats [6]. In a census performed in 2011–2012, 85,454 Swiss goats were serologically tested for their SRLV status. The results were very encouraging, showing that only 47 goats from 41 flocks were seropositive for the SRLV-B genotype, phylogenetically clustering close to the prototypic virulent CAEV-CO [7,8]. These animals were culled and eliminated from the population. In comparison, ten times more goats were likely to be infected with SRLV-A subtypes close to the MVV strains [8]. Since CAE clinical signs were not detected, we postulated that the circulating SRLV-A strains were of low virulence despite showing an apparent tropism for the mammary gland [9,10]. Under the CAE control program, SRLV-A-infected goats did not have to be mandatorily eliminated, but it was up to the cantonal authorities to decide if the livestock owners had to slaughter the animals or were allowed to keep them in the flock. Not removing SRLV-A-positive animals from the population is an apparent vulnerability, potentially favoring the spread of these genotypes. While the annual systematic sampling for SRLV has meanwhile been discontinued, a random sample was still regularly tested for SRLV between 2013 and 2018 as part of the national monitoring and surveillance program. While this sampling focused on detecting SRLV-B subtypes, SRLV-A-positive animals were also reported and recorded. The above-mentioned policy of not mandatorily eliminating SRLV-A-positive goats was maintained.

This study aimed to investigate the transmissibility and virulence of SRLV-A subtypes in goats, to examine the clinical consequences for animals infected with these viruses, and to describe the disease frequency of these subtypes in the Swiss goat population. Based on the results, implications for future SRLV surveillance should be derived.

## 2. Materials and Methods

The study design was straightforward. We visited historically SRLV-A-positive flocks and attempted to reconstruct their history by filling out a dedicated questionnaire during an interview with the goat owner. Subsequently, a careful clinical examination was performed, and blood samples were collected for serological and nested real-time PCR (qPCR).

### 2.1. Study Population

The selection of flocks was based on historical data from the National Laboratory Information System (aRes). The pre-selection included all flocks with SRLV-A-positive goats in the 2011–2012 census and the follow-up sampling between October 2013 and April 2021 [11]. In the 2011–2012 census, 473 goats from 296 flocks tested positive for SRLV-A subtypes [8]. Details of the comprehensive analysis of the aRes data are reported in the Section 3. Overall, 18 farms tested positive for SRLV-A in both the 2011–2012 census and the follow-up testing from 2013 to 2021; therefore, these 18 farms were of main interest to this study. Using data from the Agricultural Policy Information System (AGIS), farms that tested positive but had ceased goat farming by 2021 (*n* = 5) were removed from the pre-sample. Thus, 13 potential farms remained to be enrolled in this study, and eventually, 6 farmers agreed to participate in the field survey. The non-participating goat breeders could not be convinced or coerced into participating in this strictly voluntary study. The final sample comprised 145 goats from 6 flocks, and farm visits were carried out between June 2022 and October 2022. All the animals from the participating flocks were sampled and examined. Most flocks were dairy flocks except for flock #2, a suckler goat herd, and flock #5, whose owner keeps the goats for pasture care. The average flock size was 24 goats, with the smallest flock of 13 goats and the largest of 32.

### 2.2. Diagnostic Tests

#### 2.2.1. Serology

Serological analyses were performed using tools routinely used by the Swiss SRLV reference laboratory [6,12]. Sera were screened with the CAEV/MVV Total Ab Screening Test, abbreviated as “Chekit” in this manuscript (Chekit, Idexx Laboratories, Liebefeld, Switzerland), and the Caprine Arthritis–Encephalitis Virus Antibody Test Kit, cELISA (VMRD Inc., Pullman, WA, USA). Subsequently, sera were tested on selected SU5 peptides to determine the genotype of the infecting virus [13,14]. The following five peptides were used: SU5-615, derived from a Swiss SRLV-B1 field isolate; SU5-1163, derived from a Swiss SRLV-A4 field isolate; SU5-IS, derived from the prototypic SRLV-A1 Icelandic isolate 1514, SU5-Zn643 derived from a Swiss sheep SRLV-A3 isolate, and SU5-isol3, derived from a Polish SRLV-A2 isolate. For the SU5 ELISAs, we used a panel of well-characterized SU5 subtype-specific historical sera as a positive control and the serum of a certified SRLV-negative goat as a negative control.

#### 2.2.2. Real-Time PCR

In parallel to the serological analyses, DNA was extracted from buffy coat cells using the Qiagen DNeasy^®^ Blood and Tissue kit (Qiagen, Hilden, Germany), and nested real-time PCR (qPCR) was carried out in two successive amplification steps as described previously [12]. DNA extracted from infected cell cultures was used as a positive control, while DNA extracted from uninfected cultures was the negative control. The strains used to infect LSM and GSM, respectively, were MVV ATCC VR-779 and CAEV ATCC VR-905.

Positivity criteria according to the Swiss SRLV reference laboratory were as follows: Only goats serologically positive in both Chekit and cELISA and/or positive in qPCR are considered positive. A goat is considered SRLV-negative if this condition is not fulfilled (e.g., only Chekit positive). Additionally, positive results from SU5 ELISA do not determine the serological status but provide evidence of the subtype of virus circulating in the flock.

### 2.3. Clinical Examination

The main focus of the clinical examination was centered on typical signs of lentiviral disease, such as mastitis, swollen or tender joints, and neurological symptoms. Furthermore, nutritional status, posture, behavior, and general well-being were assessed. Because of organizational efficiency, rectal temperature was not measured. The lungs were auscultated, and nasal discharge, if present, was documented. Udders were examined and palpated for any signs of mastitis. Oral mucosal membranes and sclera were evaluated, obtaining evidence of possible parasitosis. The lymph nodes were also assessed to see whether they were enlarged. The same veterinarian (F.V.) carried out all clinical examinations on the 6 farms enrolled in the field survey.

### 2.4. Interviews and Questionnaire

The veterinarian in charge (F.V.), who also carried out the clinical examinations, interviewed the goat owners using a questionnaire that was filled out during the interview. The survey included collecting general information about the farm and questions about breeding, animal movements, purchases, contacts with sheep and goats from other farms, and the handling of SRLV-positive goats. The questionnaire is provided in the Appendix A.

## 3. Results

### 3.1. Analysis of Historical Laboratory Data

The analysis of historical laboratory data (aRes) comprised two time periods: data from the 2011–2012 census (period 1) and data from the 2013–2021 monitoring phase (period 2). The analysis revealed that of the 296 farms that tested positive for SRLV-A in the 2011–2012 census, 159 farms (54%) had been retested in period 2. Of these 159 farms, 18 (11%) tested positive, and 141 (89%) tested negative for SRLV-A. In period 2, 330 goats from 124 farms tested positive for SRLV-A. Of the 18 overlapping farms that tested positive in both periods, 979 goats had been tested in period 2, of which 57 goats tested SRLV-A positive (5.8%). Consequently, these 18 farms were accountable for 17% (57 of 330) of the positive animals tested in period 2. In period 1, these farms accounted for 50 SRLV-A-positive goats (11% of 473).

### 3.2. Serology and Real-Time PCR

In the framework of this study, a total of 145 goats from six flocks were tested using the standard diagnostic tests described above. The results are shown in Table 1 and Figure 1. Eight goats, seven of which were from the same flock (flock #5), were positive for an SRLV-A virus, according to the Swiss SRLV reference center criteria (see Section 2 and Figure 1).

Five goats, four from the seropositive flock #5, were positive in the qPCR for the SRLV-A genotype. Sixteen additional goats were detected as positive using a panel of experimental SU5 peptides, encompassing SU5 regions of the SRLV-A but not the SRLV-B genotype. All serological and qPCR results of the participating flocks are shown in Figure 1. Below, we describe the test results for each flock in detail.

Flock #1 (*n* = 21) had only one seropositive animal in Chekit. This serum was negative in cELISA and positive in three SU5-A (SRLV-A2, -A3, -A4) peptides. Two additional goats were positive in the SU5-A ELISA. This flock was negative in qPCR.

Flock #2 (*n* = 30) was serologically negative in Chekit and cELISA. One goat of this flock was positive in qPCR for the SRLV-A genotype but negative in all serological tests. One goat was strongly positive in the SU5-A4 and positive in the SU5-A2 ELISA, and one goat was positive in the SU5-A2.

Flock #3 (*n* = 22) was negative in both standard serological tests and qPCR, but eight goats were positive on different SRLV-A (A1, A2, A4) but not -B-derived peptides.

Flock #4 (*n* = 27) was negative in all tests used by the reference laboratory, but one animal was weakly positive in the SU5-A2 ELISA.

Flock #5 (*n* = 13) had eight positive animals in the Chekit and five in c-ELISA. Four goats were positive in qPCR for the SRLV-A genotype with Ct values of 11, 22, and in two animals with a value of 26. Seven of the eight Chekit-positive goats were also positive with at least one SRLV-A but not SRLV-B SU5 peptides.

Flock #6 (*n* = 32) was negative in all the routine tests, and only one animal was positive in the SU5-A2 ELISA.

### 3.3. Clinical Examination

The clinical examination of all animals did not reveal any key symptoms indicating SRLV infection. The udders were healthy on palpation, and no goats showed signs of carpitis. The lungs also showed no signs of disease. The animals of Flock #1 had pale mucosal membranes, shaggy coats, and mild signs of diarrhea, pointing to parasitic infections. Flock #2 had a history of moderate parasitosis. The goats in Flocks #3 and #5 showed no signs of disease. Flock #4 had one animal with diarrhea; three were under antibiotic treatment for suspected *Coxiella* infection. Two goats of Flock #6 had swollen mandibular lymph nodes. On this farm, some animals had previously been diagnosed with paratuberculosis and were euthanized.

### 3.4. Animal Movements and Potential Transmission Routes

Following the same subdivision in flocks as above, the owners reconstructed the history of their flocks as follows. An overview is presented in Table 2.

Flock #1: The goats are sporadically in contact with sheep of unknown SRLV status, and few animals are purchased from other farms every year. In 2011, there were only four goats in the flock. The SRLV-A-positive animals remained in the flock at the time of the census, and their offspring were not serologically tested.

Flock #2: After the census, a few animals were purchased in Switzerland, and these animals entered the flock without testing. The animals of this flock are never in contact with other goats or sheep. In 2011, the flock comprised about 50 goats. The SRLV-A-positive animals were eliminated at the time of the census and had no offspring.

Flock #3: This dairy flock had 22 goats at the time of the survey in 2022, but had about 50 goats in 2011. The flock is autarkic and does not purchase animals from other flocks. The positive animals at the time of the census remained in the flock.

Flock #4: At the time of the census, flock size was similar (*n* = 25). The positive animals were eliminated, their offspring remained in the flock, and they were not serologically tested. One male is purchased from another farm every second year without serological testing. These animals have no contact with other goats or sheep.

Flock #5: These goats are kept for pasture care, and the main activity is sheep meat production. In the wintertime, the owner keeps about 200 sheep separated from his goats. However, during summer, sheep and goats are moved together to an alpine pasture, together with an additional 300 sheep from two other owners. Random samples are regularly taken from these sheep and tested for SRLV antibody. The samples were always negative; however, the farmer could not remember the number of sheep tested and the frequency of these tests. At the time of the census, he owned about 30 goats; the positive animals remained in the flock, and the last one was eliminated in 2022.

Flock #6: The only purchases from external flocks concern the introduction of a new male every second year. Flock size was about the same as in 2011. Due to a change in ownership, the fate of the positive animals at the time of the census is unknown.

## 4. Discussion

The idea behind this study was to put us in the most propitious situation to discover potential problems derived from a national eradication campaign focused exclusively on the SRLV-B genotypes, leaving the SRLV-A strains almost unchallenged. This campaign effectively eliminated the virulent SRLV-B genotype and the clinical manifestations of SRLV infections in Swiss goats [6]. Only animals positive for SRLV-B and their offspring were compulsorily eliminated. In contrast, the fate of the SRLV-A-positive animals was left in the hands of the cantonal authorities and the owners, respectively. Notwithstanding a long and challenging national eradication campaign, it was surprising that the owners did, to different extents, accept the risk of keeping or introducing infectious animals in their flocks. In fact, none of the breeders serologically tested the animals introduced into their flocks.

Despite the study limitations that will be addressed below, the results of this study are somewhat surprising but, at the same time, encouraging. Only one out of six owners eliminated the SRLV-A seropositive animals in 2011, while the others either kept the seropositive goats (three flocks) or eliminated the positive adult animals, keeping their untested offspring (one flock). In the last flock, the fate of the seropositive animals is unknown. The interviews revealed that none of the farms were completely free from the three common risk factors: purchases of untested animals, contact with sheep and goats from other farms, and failure to eliminate SRLV-A-positive goats or their offspring [15]. The results are surprising because more than 10 years after the census, except for flock #5 (7 of 13 positive animals), only one SRLV-A real-time PCR-positive animal was detected. A herd containing positive animals left without taking control measures for more than ten years appears to offer the perfect opportunity to measure the virus’s ability to spread in a group of predominantly seronegative animals. The fact that this did not occur, except for one flock, is strong evidence that these SRLV-A subtypes cannot be transmitted efficiently in a flock despite the close contact between seropositive and negative animals. This agrees with the observation that the lactogenic transmission of SRLV-A in goats is less efficient than that of the SRLV-B genotype [16]. This is, however, not a general rule. In goats coinfected with SRLV-A10 and SRLV-B1, the former virus was transmitted more efficiently to the suckling kits, pointing to significant differences between genotypes [17]. In SRLV-infected sheep, horizontal transmission plays a crucial role in maintaining an infectious chain in a flock, while maternal transmission is considered irrelevant [18]. Therefore, we are tempted to conclude that the horizontal and lactogenic transmission of certain SRLV-A strains in goats is inefficient.

The SU5 ELISA results should not be overinterpreted, but clearly point to infections restricted to the SRLV-A genotype. The presence of several animals positive for the SRLV-A2 subtype in flock #5 and in three additional flocks was surprising because this particular subtype is predominant in North America, and we added this peptide, derived from a Polish SRLV-A2 isolate, as a control [19,20]. Without sequence data from SRLV infecting these animals, we cannot exclude the introduction of this SRLV subtype in these flocks. However, the most likely explanation is that this SRLV-A2 SU5 peptide encompasses a highly cross-reactive epitope recognized by antibodies induced by different SRLV subtypes. We will test this hypothesis using a collection of historical sera. Seropositive animals in SU5-ELISAs in flocks that were negative in all the other tests may be due to false-positive results. This is, however, unlikely, because the included SU5-B peptide may be considered an internal negative control for unspecific reactions. We think that the high sensitivity of the SU5 ELISAs, conferred by the immunodominant nature of these peptides [21], explains these results and may detect animals capable of controlling or even eliminating the infecting virus as previously described [22].

Unsurprisingly, the goats of the seronegative flocks did not show clinical signs of SRLV infection. Still, it is encouraging that no clinical manifestations were detected in the high-prevalence flock, particularly in the mammary gland, which was shown to be a privileged target for infections with SRLV-A4 [9]. This is strong evidence of the attenuated nature of these viruses.

The circulation of these seemingly less virulent SRLV-A strains resembles the situation in Roccaverano goats, where the avirulent genotype SRLV-E circulates uncontrolled. When infected by virulent SRLV-B, goats positive for SRLV-E do not show evident pathologies and appear to control the SRLV-B proviral load efficiently [23]. Therefore, SRLV-E has been proposed as a natural vaccine that should not be eradicated and might protect goats from the consequences of an SRLV-B infection [24,25]. No data support the notion that the SRLV-A genotype may have a similar protective effect. Their limited transmission efficiency would likely limit this effect, but it would undoubtedly be worth evaluating this possibility. If this were the case, the Swiss eradication campaign aimed solely at eliminating SRLV-B could prove to be a positive coincidence in case of an unfortunate reintroduction of the SRLV-B genotype.

The high seroprevalence in flock #5 and the fact that these animals graze together with sheep in the summertime point to a potential danger of transmission between these goats and the sheep, possibly by contamination of the pasture with virus-containing milk [26]. This may be relevant for wild small ruminants sharing these pastures as well. The literature describes several instances of cross-species transmissions between domesticated and wild small ruminants involving different SRLV genotypes [27,28,29], and infected sheep were postulated to be a crucial vector for the expansion of these viruses [30]. Sheep infected with SRLV-A4 are very efficient in transmitting the virus to goats [13], and transmission of this subtype from goats to sheep was also described [31]. Horizontal infections are the principal route of virus transmission in sheep, while in goats, lactogenic transmission may predominate [18,32,33,34,35]. The sheep in contact with flock #5 were regularly tested for SRLV infections by random sampling of a subset of animals and were negative. Unfortunately, the farmer did not know the details of the testing procedure on the sheep he hosted during the summertime. This notwithstanding, this suggests that horizontal infections between infected goats and sheep are not frequent with the SRLV subtypes involved, particularly in the context of extensive farming, which was shown to prevent SRLV infections in sheep [36].

As for the lactogenic transmission mentioned above, the subtypes involved may play a crucial role in the efficiency of horizontal transmission, as shown by different studies focusing on SRLV transmission in mixed flocks of sheep and goats [3,37,38]. In fact, it would be important to characterize the SRLV circulating in flock #5. The ‘hosts’ traits also influence the efficiency of infection, immune reactions, and induced sequelae, potentially contributing to the absence of clinical signs in these goats [34,39,40,41].

Can these six flocks be considered representative of Switzerland? For many years, SRLV-induced carpitis, once a widespread disease in Swiss goats, has no longer been detected in Switzerland. SRLV-A strains were detected during the census in 2011–2012 [8]; they were not systematically eradicated and are still circulating in our country. Therefore, we are convinced that these flocks, selected according to their SRLV status in 2011–2012, are representative and illustrate the current SRLV situation in Switzerland. Despite the limited sample size, it can be assumed that 46% of SRLV-A-positive farms (*n* = 137) that were not retested after the census currently keep SRLV-A-positive animals in their flocks. When assuming a similar between-herd and within-herd prevalence, they have approximately 50 SRLV-A-infected goats. In addition, the slight increase (from 11% to 17%) in the proportion of SRLV-A-positive animals from the 18 farms that tested positive both during the census and between 2013 and 2021 suggests that historically, SRLV-positive goat farms also pose a higher risk of disease transmission for the SRLV-A genotypes.

The most apparent limitation of this work is the restricted number of flocks and animals involved. One reason can undoubtedly be attributed to farmers’ reluctance to participate in this type of study. The long and laborious eradication campaign launched in the 1980s, regularly accompanied by frustrating setbacks with heavy consequences for breeders, has not fostered enthusiasm to participate in this study.

Another limitation is that we have refrained from isolating and characterizing the viruses involved in molecular detail, which was dictated by limited time and personnel resources, certainly not by a lack of interest.

## 5. Conclusions and Recommendations

Based on the study outcomes, future SRLV monitoring in Switzerland should focus on two aims: to maintain the SRLV-B-negative status and to monitor for potential changes in the virulence and transmissibility of the circulating SRLV-A genotypes. Considering the very low seroprevalence of SRLV-B and SRLV-A strains, a serological survey requires a large sample size, which can be resource-intensive and challenging to achieve. Combining SRLV surveillance with other serological surveys or surveillance programs (e.g., brucellosis) might also not be ideal, as sampling should be risk-based and focus specifically on historically positive farms. Clinical monitoring should be emphasized for the appearance of symptoms related to SRLV infections, such as indurative mastitis and carpitis. Therefore, veterinarians and breeders should maintain and foster disease awareness. Even though this strategy is not without risks due to the subtle development of the clinical symptoms, it was shown to work in one case of import of SRLV-B1-infected animals where the veterinarian in charge immediately recognized the arthritic symptoms in some of the imported animals, linking them to a potential SRLV infection, later confirmed by laboratory analysis, and avoiding the reintroduction of SRLV-B1 in the Swiss goat population [6].

Therefore, we are convinced that targeted serological and virological analyses, combined with maintained disease awareness, will permit the efficient control of potential SRLV-B infections and the resurgence of SRLV-induced pathologies.

## Figures and Tables

**Figure 1 pathogens-14-00719-f001:**
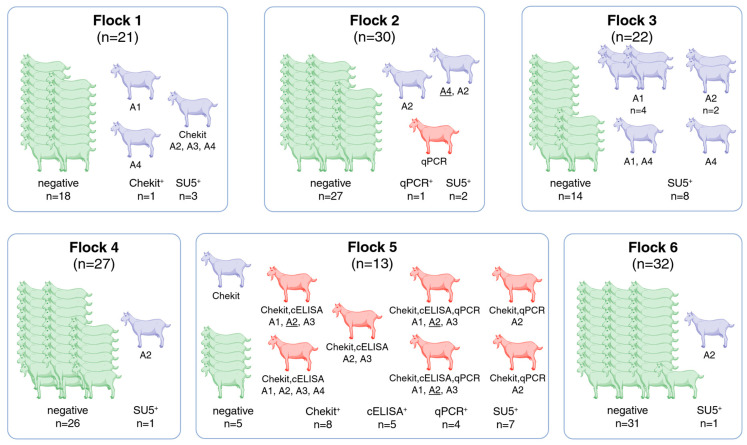
All results from the four diagnostic tests applied: (1) Chekit, (2) cELISA, (3) qPCR, and (4) SU5. Goats depicted in green gave negative results in all tests; those in red are positive according to the Swiss SRLV reference center criteria *, while those in purple are positive in at least one test but are not considered positive by these criteria. SU5 serological results (comprising the following SU5 peptides: A1, A2, A3, and A4) do not determine the serological status but provide evidence of the subtype of virus circulating in the flock. In the case of reactions with multiple peptides, the dominant peptide, if applicable, is underlined. Reported numbers may contain multiple counts of individual animals if they have tested positive in several tests. * Swiss SRLV reference center criteria: only goats serologically positive in both Chekit and cELISA and/or positive in qPCR are considered SRLV-positive.

**Table 1 pathogens-14-00719-t001:** Summary of the serological and qPCR results *.

	Flock 1	Flock 2	Flock 3	Flock 4	Flock 5	Flock 6	Total
No. goats	21	30	22	27	13	32	145
Tested negative	18	27	14	26	5	31	122 (84%)
SRLV-positive by reference lab definition (red in Figure 1)	0	1	0	0	7	0	8 (5.5%)
SRLV-positive due to positive test results, but officially SRLV negative (yellow in Figure 1)	3	2	8	1	1	1	15 (10.3%)
Chekit	1				7		8 (5.5%)
cELISA					5		5 (3.4%)
qPCR		1			4		4 (2.8%)
SU5 A1	1		5		4		10 (6.9%)
SU5 A2	1	2	2	1	7	1	13 (9.0%)
SU5 A3	1				5		6 (4.1%)
SU5 A4	2	1	2		1		6 (4.1%)

* An individual goat can test positive by multiple tests, see Figure 1.

**Table 2 pathogens-14-00719-t002:** Summary of the questionnaire and interviews with the goat owners.

	Flock 1	Flock 2	Flock 3	Flock 4	Flock 5	Flock 6
No. goats	21	30	22	27	13	32
No. females	20	21	21	26	13	32
No. males	1	9	1	1	0	0
Production type	dairy	suckler	dairy	dairy	pasture care	dairy
Member SZZV *	yes	no	no	yes	yes	yes
Contact with sheep or goats from other flocks	yes (sheep sporadic)	no	no	no	Yes (>300 sheep)	unknown
Buy-in of animals	yes	yes	no	yes	unknown	yes
SRLV-positive animals or offsprings remained in herd	yes	no	yes	yes (offspring)	yes	unknown

* SZZV: Swiss Goat Breeding Association.

## Data Availability

All data are summarized in Table 2 and Figure 1. The raw data can be requested from G.B.

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
