# Peer review of "A Look Under the Carpet of a Successful Eradication Campaign Against Small Ruminant Lentiviruses"

_pathogens, 2025, doi:10.3390/pathogens14070719_

Round 1
Reviewer 1 Report
Comments and Suggestions for Authors
A look under the carpet of a successful eradication campaign 2 against small ruminant lentiviruses
This is an interesting article which revisits the issues with Lentiviral infections in small ruminants in Switzerland. Using previous data the team visit farms with a known history of infection and check the animals here for clinical signs and seroconversion as well as active viral carriage, and interview the farmers to obtain additional information.
The study is an interesting aspect on surveillance and adds to the knowledge in the field, as well as in the country. It suffers from somewhat low numbers, but this is an expensive study and labour intensive so that is to be expected.
The study is well carried out, has some interesting data included and some useful conclusions.
I feel that some additional data in the methodology from the previous studies maybe useful, but there is a list of some suggestions below.
Line 43- perhaps reword this as it is a little unclear
Methods- I think it is worth including some data on the blood sampling, if all animals in a flock were tested etc
Also worth a mention of controls used for the ELISAS and PCR assays. I guess that these are as per previous studies but worth stating I feel
You seem to jump between calling them flocks and herds- please be consistent.
Line 180- please italicise Coxiella.
Figure 1- it is difficult to tell the difference between red and orange, although may just be my copy. Is it possible to make these brighter or alternate colours?
Line 266- is there a reason why this is the case?
Author Response
Comments 1: This is an interesting article which revisits the issues with Lentiviral infections in small ruminants in Switzerland. Using previous data the team visit farms with a known history of infection and check the animals here for clinical signs and seroconversion as well as active viral carriage, and interview the farmers to obtain additional information.
The study is an interesting aspect on surveillance and adds to the knowledge in the field, as well as in the country. It suffers from somewhat low numbers, but this is an expensive study and labour intensive so that is to be expected.
The study is well carried out, has some interesting data included and some useful conclusions.
Response 1: We thank the reviewer for his appreciation of the study and the difficulties inherent in such field studies.
Comments 2: I feel that some additional data in the methodology from the previous studies maybe useful, but there is a list of some suggestions below.
Line 43- perhaps reword this as it is a little unclear
Response 2: Comments 3: We agree with the reviewer that this sentence may be unclear. Here is the revised sentence (Lines 43-45 in red): In this respect, Switzerland paid the price for pioneering an eradication campaign in goats before the potential risk of SRLV transmission between sheep, excluded by this campaign, and goats was known.
Comments 3: Methods- I think it is worth including some data on the blood sampling, if all animals in a flock were tested etc
Response 3: We agree and added this sentence to paragraph 2.1, lines 90-91, in red:
All the animals from the participating flocks were sampled and examined.
Comments 4: Also worth a mention of controls used for the ELISAS and PCR assays. I guess that these are as per previous studies but worth stating I feel
Response 4: Thank you for your suggestion. We added in the material and methods section these missing controls.
Lines 112-114 in red: DNA extracted from infected cell cultures was used as positive control, while DNA extracted from uninfected cultures was the negative control. The strains used to infect LSM and GSM, respectively, were MVV ATCC VR-779 and CAEV ATCC VR-905.
ELISA, lines 105-107 in red: For the SU5 ELISA, we used a panel of well-characterized SU5 subtype-specific historical sera as a positive control and the serum of a certified SRLV-negative goat as a negative control.
Comments 5: You seem to jump between calling them flocks and herds- please be consistent.
Response 5: We agree with our reviewer and have corrected this inconsistency by using “flock” throughout the manuscript. Lines: 23, 25, 77, 78, 90, 158, 334, and Table 1
Comments 6: Line 180- please italicise Coxiella.
Response 6: Thank you for pointing this out; we have italicized Coxiella (red, line 185).
Comments 7: Figure 1- it is difficult to tell the difference between red and orange, although may just be my copy. Is it possible to make these brighter or alternate colours?
Response 7: Thank you very much for this valuable input! We adjusted the colors and saturation for better readability. The “orange goats” are now purple and the legend has been modified accordingly.
Comments 8: Line 266- is there a reason why this is the case?
Response 8: We prefer not to speculate in the absence of concrete data. However, one possible explanation is that the highly efficient horizontal transmission of SRLV-A4 between infected sheep and negative goats (see reference 13 and answers to questions 3 and 4 of reviewer #2) may be due to the tropism of these viruses for the sheep's lungs and the ensuing efficient horizontal transmission via respiratory secretions. This is not the case in goats (ref. 9), which may not transmit SRLV-A efficiently through horizontal contact. Notably, the goats did not transmit the virus efficiently via colostrum, despite the high viral loads detected in SRLV-A4-infected goats in the mammary glands (ref. 9). It is possible that the virus circulating in flock #5 did not exhibit this marked mammary gland tropism characteristic of the SRLV-A4 subtype. Still, unfortunately, we don't have information on this point.
Reviewer 2 Report
Comments and Suggestions for Authors
- The study visits only 6 out of the 18 historically SRLV‑A–positive farms (lines 82–87). How were these six chosen, and how do they compare (in geography, flock management, breed) to the 12 that declined participation? Could there be systematic differences that affect the transmissibility findings?
- The manuscript asserts that these six flocks “are representative and illustrate the current SRLV situation in Switzerland” (lines 321–326), yet no formal sampling calculation or power analysis is provided. Please include an a priori or post hoc power assessment to justify the sample size and support claims of representativeness.
- The use of SU5 peptide ELISAs (SU5‑A1–A4; SU5‑B) is central to subtype determination (lines 98–102; 267–275). Could the authors provide sensitivity/specificity data for each SU5 assay in Swiss field conditions? Have these assays been validated against sequence-confirmed isolates from Switzerland?
- In multiple flocks, goats were SU5‑positive but negative by Chekit, cELISA, and qPCR (e.g. Flocks 1, 3, 4, 6; Table 2). Can the authors clarify whether these are likely false positives, low‐level infections, or evidence of viral clearance? Would additional confirmatory testing (e.g. western blot, virus isolation) help resolve their status?
- No sequence data are presented for the SRLV‑A strains detected. Given the unexpected SU5‑A2 reactivity (lines 268–275), can the authors include (or plan to include) proviral sequencing (e.g. gag, pol or env regions) to:
-
-
Confirm the circulating SRLV‑A subtypes,
-
Assess genetic divergence from reference strains, and
-
Evaluate potential recombination events.
-
If sequencing is not yet available, this limitation should be explicitly acknowledged in the Discussion, and plans for future characterization outlined.
-
Author Response
Comments 1: The study visits only 6 out of the 18 historically SRLVA–positive farms (lines 82–87). How were these six chosen, and how do they compare (in geography, flock management, breed) to the 12 that declined participation? Could there be systematic differences that affect the transmissibility findings?
Response 1: We agree with the reviewer that this is an explicit limitation of our work. In discussing our manuscript (lines 321–326), we stated that the farmers, who paid a high tribute to a lengthy and arduous eradication campaign, were, euphemistically speaking, "reluctant" to participate in this study. The participation was, by definition, on a voluntary basis, and we had no means at our disposal to force the breeders to participate. The following sentence was added to the manuscript (lines 87-88, red.): The non-participating goat breeders could not be convinced or coerced into participating in this strictly voluntary study.
Thank you also for the important question regarding potential selection bias and how this might have influenced our study outcomes. Of the 18 historically SRLVA–positive farms, 5 have given up goat farming and could therefore not contribute to the study, while 7 farms declined participation. We do not know management practices or breeds of these farms. However, we have information about the location and the presence of sheep. Geographically, all seven excluded farms come from a region that is represented in the sample through a participation farm. Five of the seven farms do not have sheep present, while two farms have sheep. This aligns well with the representation in the sample. As the transmission is strongly influenced by the management practices (e.g. contact to (SRLV-positive) goats or sheep from other herds, buy-in of animals or management/culling of SRLV-positive animals) it is difficult to quantify the effect of this uncertainty. Overall, we believe that the heterogeneity of our sample also reflects the excluded farms relatively well, and therefore the exclusion of these farms does not substantially affect our study outcomes.
Comments 2: The manuscript asserts that these six flocks “are representative and illustrate the current SRLV situation in Switzerland” (lines 321–326), yet no formal sampling calculation or power analysis is provided. Please include an a priori or post hoc power assessment to justify the sample size and support claims of representativeness.
Response 2: Thank you for your comment. We have discussed the origin and composition of the sample above and would like to emphasize that the number of farms and goats included in the study is based on all available study subjects, as we have tried to obtain as much information as possible. The study pursued various objectives, namely serological, virological, and clinical examinations on animal and herd level. In total, we had 145 animals in our sample available and an average herd size of 24 goats. To detect an estimated prevalence of 5%, with 0.05 precision and 0.95 confidence level, 49 animals would have needed to be sampled. To detect one positive goat in a herd of 24 goats, the required sample size is 18 goats (with 0.05 precision and 0.95 confidence level). We have tested all animals within each herd. We are aware that the small sample size of our study is insufficient to demonstrate freedom from disease or representative within-herd prevalence estimates for the population. However, it allows to draw general conclusions about the transmissibility of SRLV in Swiss goat herds under the current epidemiological situation.
Comments 3: The use of SU5 peptide ELISAs (SU5A1–A4; SU5B) is central to subtype determination (lines 98–102; 267–275). Could the authors provide sensitivity/specificity data for each SU5 assay in Swiss field conditions? Have these assays been validated against sequence-confirmed isolates from Switzerland?
Comments 4: In multiple flocks, goats were SU5positive but negative by Chekit, cELISA, and qPCR (e.g. Flocks 1, 3, 4, 6; Table 2). Can the authors clarify whether these are likely false positives, low‐level infections, or evidence of viral clearance? Would additional confirmatory testing (e.g. western blot, virus isolation) help resolve their status?
Responses to comments 3 and 4: To address the excellent and well-justified questions 3 and 4, we provide a brief summary of the results obtained with these peptides. In the original publication by Mordasini et al. (ref. 14), we demonstrated that the antibody response to this antigenic region of Env is robust and immunodominant, inducing high-avidity antibodies in infected animals. The short SU5 peptide contains at least two epitopes: one is relatively conserved across different genotypes, and the other is highly variable. Only 4 out of 81 PCR-positive and WB-positive samples (5%) reacted with the SU5 constant region, while 51 (63%) recognized the variable region. Using the complete SU5 peptide, encompassing both regions (25-26 amino acids, depending on the strain), 77 samples were positive (95%). This number reflects the sensitivity of the SU5 assay when using peptides that fit the circulating strains.
In the same publication, we showed that in a field case, using genotype-specific SU5 peptides, we could predict the genotype of the infecting virus as determined by sequencing. In this case, we have already seen that the type-specific reaction is not purely type-specific, and cross-reacting antibodies are detected in several animals.
The best illustration of these complex patterns of reactivity is shown in our paper reporting a horizontal infection between sheep and goats performed under field conditions by co-housing nine seronegative goats with 14 SRLV-A4 infected sheep (ref. 13). As shown in Fig. 1 of this report, three goats from our certified CAEV negative flock, created by separating the newborns at birth from their seronegative mothers and raising them with cow colostrum, were perfectly negative on all SU5 peptides at day 0. At week 12, they all seroconverted to the specific SU5-A4 peptide, and at week 93, they were all positive with the SU5 peptide corresponding to the infecting SRLV-A4 virus, as well as with the SU5-A1 peptide. None of the animals showed a positive response to the SRLV-B SU5 peptides, and only one out of three animals seroconverted in the routinely used ELISA (IDEXX CAEV/MVV Total Ab Test, Idexx Laboratories, Liebefeld, Switzerland; a whole-virus antigen based indirect ELISA). However, all three were PCR positive, and the sequenced amplicons confirmed an SRLV-A4 infection.
In contrast, the six seronegative goats born to SRLV-B1-infected mothers, which never seroconverted in the routinely used ELISA or WB, and were all PCR-negative, showed a complex seroconversion to different SU5 peptides. Two of these goats were already positive with SU5-B peptides at day 0. Three of these six goats began to seroconvert in week 4 after contact with the infected sheep, and by week 12, they were all seropositive with different SU5-A and -B peptides. Only two animals were positive in the PCR test. This is compatible with the induction of a de-novo immune response to SU5-A peptides and an anamnestic response to SU5-B peptides.
Finally, in the Swiss samples tested in reference 12, by Schaer J. et al. (Table 1), the results of 23 PCR-positive samples are shown. Eighteen of these samples were positive in the SU5 ELISA (sensitivity 78%), and the SU5-ELISA classification in SRLV-A or -B genotype matched the PCR results. Of the five potentially false-negative SU5 samples, 3 were PCR-positive but negative in all ELISAs, suggesting that these animals were infected but had not yet seroconverted. One serum was positive only in one of the ELISAs used (VMRD), while the last sample can be considered a clear SU5 false-negative sample.
In conclusion, we can confirm that the SU5-ELISA has high sensitivity; however, we cannot precisely assess its specificity. This is crucial to answering the highly relevant questions of our reviewer, which were constantly on our minds. While analyzing other flocks that have shown similar results in the past, we aimed to do precisely what our reviewer is suggesting. However, we were unable to isolate a virus from animals that showed only an SU5-positive serology but were negative in all other tests.
Western blot works well with SRLV-B-infected goats but is less sensitive with SRLV-A-infected animals (ref. 12). The vast majority of SU5-peptide-only seropositive animals were negative in both Western blot and qPCR.
In this work, one qPCR-positive animal was negative in the SU5 peptides ELISA, while four SRLV-A qPCR-positive animals from flock #5 reacted with different SU5-A peptides. As stated above and in the discussion (lines 276-277), we cannot exclude the possibility that the SU5-peptide-only seropositive animals are false positives. The only indirect evidence that this may not be the case is that none of these sera reacted with the SU5-B1 peptide, which can be considered a negative control for unspecific reactions.
Ref. 13 enclosed.
Comments 5: No sequence data are presented for the SRLVA strains detected. Given the unexpected SU5A2 reactivity (lines 268–275), can the authors include (or plan to include) proviral sequencing (e.g. gag, pol or env regions) to:
Confirm the circulating SRLVA subtypes,
Assess genetic divergence from reference strains, and
Evaluate potential recombination events.
If sequencing is not yet available, this limitation should be explicitly acknowledged in the Discussion, and plans for future characterization outlined.
Response 5: These questions are perfectly on target, and unfortunately, we don't have an answer yet. Unfortunately, using the limited material available, we were unable to obtain the sequence information necessary to phylogenetically classify the virus(es) circulating in flock #5, which showed a peculiar reaction with the SU5-A2 peptide. We acknowledged this in the discussion in lines 340-342. If the opportunity arises to obtain blood samples from this flock, we will not hesitate to perform the suggested analysis. However, at present, we do not have the necessary resources, and our animal experimentation license has expired; there is no quick fix for this problem.
Notwithstanding this, we tend not to overemphasize these SU5-A2 reactions. As discussed in our answers to questions 3 and 4, the SU5 peptides help discriminate between SRLV-A and -B infections, but are less reliable in distinguishing the subtypes involved. As discussed above and in reference 13, the breadth of these more or less type-specific reactions is likely influenced by the immunological memory induced by previous encounters with different SRLV subtypes. We discussed this point in lines 267-275 and proposed testing the hypothesis that the SU5-A2 peptide contains a highly cross-reactive epitope using a historical collection of sera.

Round 2
Reviewer 2 Report
Comments and Suggestions for Authors
My comments have been addressed. Thanks!